# Validity and Reliability of an Offline Ultrasound Measurement of Bladder Base Displacement in Women

**DOI:** 10.3390/jcm11092319

**Published:** 2022-04-21

**Authors:** Sandra Martínez-Bustelo, Asunción Ferri-Morales, Fernando J. Castillo-García, Antonio Madrid, M. Amalia Jácome

**Affiliations:** 1Psychosocial Intervention and Functional Rehabilitation Research Group, Faculty of Physiotherapy, University of A Coruña, Campus de Oza, CP 15006 A Coruña, Spain; s.martinez1@udc.es; 2Faculty of Physiotherapy and Nursing, University of Castilla-La Mancha, Real Fábrica de Armas, CP 45071 Toledo, Spain; 3School of Industrial and Aerospace Engineering, University of Castilla-La Mancha, Real Fábrica de Armas s/n, CP 45071 Toledo, Spain; fernando.castillo@uclm.es; 4Department of Physiotherapy, Medicine and Biomedical Sciences, NEUROcom (Neuroscience and Motor Group), University of A Coruña, Campus de Oza, CP 15006 A Coruña, Spain; antonio.madrid@udc.es; 5Biomedical Institute of A Coruña (INHIBIC), University of A Coruña, Campus de Oza, CP 15006 A Coruña, Spain; 6Faculty of Science, University of A Coruña, CITIC (Centro de Investigación en Tecnologías de la Información y las Comunicaciones), Campus de A Zapateira, CP 15071 A Coruña, Spain; maria.amalia.jacome@udc.es

**Keywords:** pelvic floor muscles, ultrasound, MATLAB, validity, reliability, physiotherapy

## Abstract

The effect of different exercises on the position of pelvic organs in women has not been sufficiently assessed. The objective was to analyze the validity and reliability of a new two-dimensional ultrasound algorithm to measure offline the displacement of the bladder base during abdominal exercises. This algorithm could be a useful method to future studies in determine the most appropriate exercises in sports and in rehabilitative program for the pelvic floor in women. All subjects were tested by transverse transabdominal ultrasound. The measurements were conducted offline using a customized code written in MATLAB (Ecolab) for image-processing, and manually on the ultrasound monitor using electronic calipers. The agreement was assessed with a paired *t*-test, Pearson’s linear correlation coefficient (*r*), the Lin’s concordance correlation coefficient (CCC), the intraclass correlation coefficient ICC (A,2) and a Bland–Altman plot. The reliability was confirmed by the interdays intra-rater ICC coefficient. The results were that Ecolab and ultrasound transducer measures did not differ statistically (*p* = 0.246). Furthermore, both methods showed a very strong relationship, and the Ecolab demonstrated to be a valid and reliable method. We concluded that Ecolab seemed to be a valid and reliable tool to assess the effect of abdominal contractions in the female pelvic floor.

## 1. Introduction

Ultrasound (US) imaging can visualize the movement of pelvic floor structures during voluntary contractions and other tasks to investigate pelvic floor muscle activation [1]. It is particularly useful because: it is non-invasive, portable, safe and relatively inexpensive. Two modalities can be used, Transabdominal ultrasound (TAUS) modality, in which the transducer is applied suprapubically in a transversal or sagittal plane to measure bladder base (BB) movement as an indicator of pelvic floor muscles (PFM) function [2]; Transperineal ultrasound (TPUS) modality, in which case the transducer is positioned on the perineum in the sagittal plane to view pubic symphysis, bladder and urethra in order to estimate the displacement of the bladder-neck during PFM contractions [3].

As pelvic floor contraction has an effect on the pelvic organs’ position, several authors have quantified the amount of movement occurring at the BB during voluntary PFM contractions using TAUS [4,5,6]. Although TPUS modality has higher reliability during functional maneuvers, it is more invasive than TAUS [7]. Advantages of using TAUS in the general exercising population include its high-speed results, non-invasive technique, absence of the need to be undressed, and its direct visualization of pelvic floor movement during contractions [8,9]. In addition, good inter-rater reliability for the measurement of BB (transverse and sagittal view) during PFM contraction using TAUS has been reported [5,10,11].

The majority of researchers use on-screen calipers from an US device to measure the BB displacement that occurs during a PFM contraction or other maneuvers [4,5,7]. However, displacement of BB has not been yet analyzed via an offline MATLAB algorithm, which could reduce potential measurement errors when using on-screen calipers in real time. Other advantages of the MATLAB algorithm are the time saved in the clinical setting and the prevention of errors resulting from manual exportation of data to a database sheet that are commonly observed with the employment of on-screen calipers.

The study aimed to introduce a two-dimensional US algorithm to measure the BB displacement during PFM contractions offline, as well as to analyze its validity and reliability. This algorithm could be a useful method for future research to discriminate which exercises cause a descent of the pelvic organs and therefore may not be advisable in sports or in rehabilitation programs for the pelvic floor in women.

## 2. Materials and Methods

### 2.1. Participants

A convenience sample of 32 nulliparous women participated in this prospective study; 27 to calculate the validity of the MATLAB algorithm and 5 additional volunteers to determine its reliability. The inclusion criteria were to be nulliparous, willingness to participate in the study, and ability to contract PFM correctly. This ability was assessed by palpation and by superficial biofeedback electromyography (PHENIX^®®^ USB NEO, Vivaltis, Montpellier, France), reflecting the intensity and the length of the pelvic floor contraction on a monitor screen. Exclusion criteria were the inability to contract PFM properly, pregnancy, known neurological disease, or inability to understand instructions given in Spanish language. All participants gave written consent to participate and the rights of subjects were protected. This study was approved by the Galician Ethics Committee (CODE 2014/610), conformed to the Declaration of Helsinki, and was registered at ClinicalTrials.Gov PRS Protocol Registration and Results System (ID:NCT04154527).

### 2.2. Experimental Procedure

All subjects were tested by transverse TAUS while lying in a supine position with hips and knees slightly flexed and abducted, and with the lumbar spine in a neutral position. To allow clear imaging of the pelvic floor fascia, a bladder filling protocol [12] was implemented to ensure that the subjects had moderately full bladders without having an urge to urinate (less than 300 mL assessed by abdominal US using the formula described by Poston et al. 1983 [13], height x depth x width x 0.7). This protocol involved participants voiding 1 h before the assessment and then consuming 500 mL of water [2,11].

To image the pelvic floor, a 3.5 MHz curved linear array US transducer was used (LOGIQe Ultrasound, GE Healthcare, Chicago, IL, USA) with the US unit set in B mode. The same researcher, a qualified US technician, examined all the participants. Transverse TAUS of the bladder was performed via the abdominal wall by placing the probe suprapubically on the lower abdomen in a transverse plane to the linea alba. The transducer was angled at 15–30 degrees from the vertical in a caudal posterior direction to obtain a clear image of the inferior–posterior aspect of the bladder and the midline pelvic floor structures (urethra, perineal body, and rectum).

The marker to measure the displacement was situated in the middle of the BB on the junction of the hyper- and hypo-echoic areas corresponding to the deep layer of PFM [14] (see Figure 1). The BB displacements between the resting and final positions of the marker during each maneuver were then measured (see δ in Figure 1).

Subjects were instructed to randomly perform a series of four different PFM and abdominal contractions: contraction A requesting the submaximal recruitment of PFM; contractions B and C involving deep abdominal muscles (Transversus Abdominis and Obliquus Internus muscles); and contraction D involving superficial abdominal muscles (Obliquus Externus and Rectus Abdominis muscles). Table 1 shows more details about the different PFM and abdominal contractions A–D, which are depicted in column 1.

Each contraction was repeated twice and the average displacement of the BB was recorded for data analysis. Electromyography biofeedback with superficial electrodes on the perineum and lower abdominal wall recorded the submaximal contraction of PFM and deep abdominis muscles. The participants were asked to perform maximum voluntary recruitment of PFM and Transversus Abdominis muscle for normalization purposes. Subsequently, they tried submaximal contractions of both groups of muscles at 25–30% of their maximal force following the trajectory displayed on the biofeedback screen.

### 2.3. Data Processing

Analyses of the two-dimensional US displacement of the BB were conducted offline using a custom code named Ecolab written in MATLAB image-processing software (The MathWorks, Inc., Natick, MA, USA). The main goals of Ecolab were aiding in the measurement process and correctly collecting all data for further analysis.

A graphical User Interface is automatically launched upon opening Ecolab (see Figure 2). The ‘Browse’ button allows the user to select the recorded videos for analysis.

The first action for the analysis of the video consists of removing the initial and final frames of the video where there is no movement.

The recorded video is made of a total of *N* frames. Each frame has *n* rows and *m* columns that correspond with the video resolution n×m, and a third component that represents the color channel in RBG format. The value of $P_{f}\left(i,j,k\right),$ therefore, represents the color intensity of channel k of the pixel located in row i and column j of frame f, where i=1,…n;j=1,…,m, k=1,2,3 and f=1,…,N (see Figure 3).

The grayscale intensity of each pixel, $P_{f}^{g}\left(i,j\right)$, can be determined for each video frame f by calculating the mean of the RGB channel intensity:(1)$$P_{f}^{g}\left(i,j\right)=\frac{1}{3}\overset{3}{\underset{k=1}{\sum }}P_{f}\left(i,j,k\right)\;$$

Let us define the overall grayscale intensity level of each image frame *f* as:(2)$$L_{f}=\frac{1}{n\cdot m}\overset{n}{\underset{i=1}{\sum }}\overset{m}{\underset{j=1}{\sum }}P_{f}^{g}\left(i,j\right)\;\;\;$$

Once each video frame is converted to grayscale, the first video frames without movement can be easily removed by comparing its overall grayscale intensity level, $L_{f},$ to that of the following frame, $L_{f+1}$. When the difference between both consecutives frames overcomes an upper limit, say $L^{*}$, we can consider that the movement has started:(3)$$\left|L_{f+1}-L_{f}\right|>L^{*}\;$$

The upper limit $L^{*}$ can be experimentally determined by selecting two frames with an easily detectable movement between them and obtaining their corresponding [15] $\left|L_{f+1}-L_{f}\right|$.

The next step consists of determining when the posterior bladder wall presents the maximum displacement as result of the PFM contraction. A collage with the 3 × 5 most significant frames is displayed, and the user is required to select the initial frames, and final frames are required to be selected by the user. A horizontal grid overlaps the images to assist the user in the proper selection of the images (see Figure 4a).

After determining the initial and final frames with movement, a new image, one per frame, is presented to the user to select the point of interest to be measured. The Ecolab application compares the selected pixels and measures the distance converting image coordinates into real-world coordinates. The result is the vertical distance between the initial and final positions of the point of interest during the PFM contraction. This distance is shown on the graphical user interface (see Figure 4b), plotted on an additional figure and saved into an Excel file.

### 2.4. Statistical Analysis

Statistical analysis of the data was performed using SPSS 22.0 software (IBM Corporation). The validity of the MATLAB algorithm was checked by measuring the B displacement in 27 nulliparous volunteers while performing contraction A both with the Ecolab algorithm and on the US monitor using electronic calipers. The agreement between these two measurements was assessed with a paired *t*-test, Pearson’s linear correlation coefficient (*r*), Lin’s concordance correlation coefficient (CCC), and the intraclass correlation coefficient (ICC) model (A, 2) following the notation according to McGraw and Wong, 1996 [16], with 95% confidence intervals. The strength of the ICC correlation coefficient was interpreted according to Koo and Li [17], considering that less than 0.50 indicated poor reliability, 0.50 to 0.75 indicated moderate reliability, 0.75 to 0.90 indicated good reliability, and 0.90 or greater indicated excellent reliability. Finally, a Bland–Altman plot was constructed with the limits of agreement (LOA) calculated as LOA = *d* ± 1.96*SD*, where *d* is the sample mean of the differences, and *SD*, the sample standard deviation of the differences.

To check the reliability of the algorithm, the inter-day intra-rater ICC coefficient was obtained by comparing the measurements of contractions A, B, C and D in the same image in a convenience sample of 32 participants (the 27 participants included to calculate the validity of MATLAB algorithm and 5 additional volunteers) between two sessions one week apart. The significance level was set at α = 0.05 for all the outcomes.

## 3. Results

The analysis of the above results showed high validity and reliability for the use of a customized software code for the two-dimensional US measurement of BB displacement.

### 3.1. Validity of the Us Measurement

To check the validity of the MATLAB algorithm, the difference between the displacement of the BB measured via electronic calipers on a screen and with the MATLAB algorithm was calculated in 27 volunteers during contraction A (Table 2). On average, the differences between both methods were not statistically significant (*d* = 0.037, *p* = 0.246). The Pearson’s correlation coefficient (r = 0.97, *p* < 0.001) and Lin’s concordance correlation (CCC = 0.961) indicate a very strong relationship between Ecolab and US transducer measures. The ICC was also high (ICC (A,2) = 0.96, 95% CI = 0.92 to 0.98), further indicating excellent agreement between both methods. Figure 5 displays the scatterplot of the values obtained with both approaches. 

The regression fit (solid line, R^2^ = 94.3%) fell close to the 45° line (dashed line), demonstrating that both measurements tented to give yield very similar results. Figure 6 displays the Bland–Altman plot with the relevant limits of agreement LOA = (−0.35, 0.28). A total of 96.3% of the results fell within the 95% CI of the mean difference between the methods.

### 3.2. Reliability of the Ultrasound Measurement

In terms of reliability of the MATLAB calculations conducted in a sample of 32 women, the algorithm estimated an intra-rater coefficient of ICC (1,2) > 0.95 from the same image in all four contractions A, B, C and D (Table 2), showing excellent reliability of the Ecolab algorithm (ICC (1,2) = 0.96, 0.98, 0.99 and 0.98 during contractions A, B, C and D, respectively).

## 4. Discussion

Former research has studied which exercises appeared to be appropriate to both nulliparous and parous women using ultrasound imaging of the pelvic floor [2,12,18]. However, to our knowledge this is the first study to assess the validity and reliability of a MATLAB algorithm to measure the effect of different exercises on the pelvic floor in women. The present study showed that this offline methodology could be a valid and reliable tool.

A former study reported good agreement between a manual approach and a method using a MATLAB algorithm to measure the gastrocnemius fascicle length during gait, with values of multiple correlation coefficient about 0.90 ± 0.09, 95% CI = (0.86, 0.95) [19]. The repeatability of the algorithm was also high, with an overall coefficient of 0.88 ± 0.08, 95% CI = (0.79, 0.96). Other studies found the inter-day reliability to be very good using a MATLAB algorithm to measure the inter-rectus distance at rest 2 cm over the umbilicus (ICC of 0.87 (95% CI = 0.73, 0.94)) [20], or to measure the thickness of the vastus lateralis muscle (ICC 0.96 ± 0.01) [21]. However, the Ecolab represents a novel means to measure the BB displacement during different exercises or functional activities.

### 4.1. Validation of the Two-Dimensional Measurement Using Customized Code

The majority of previous studies that evaluated the BB displacement through the abdominal wall used electronic calipers on the US monitor [4,11,22]. Our research group developed a novel MATLAB algorithm named Ecolab to measure this movement that aimed to save time in the clinical setting. High agreement (ICC = 0.96, 95% CI = 0.92 to 0.98) was found between the former method (manual US) and the Ecolab application, as well as good precision for the latter.

### 4.2. Reliability of the Two-Dimensional Measurement Using Customized Code

To study the reliability of this novel US measuring tool, several potential sources of measurement errors were considered: the subjects, the testing, the scoring, the instrumentation and factors such as the instructions from the examiner [23]. To mitigate these potential errors, the position of the subject, the examiner’s instructions to the participants, the transducer location and inclination, and the position of the marker to measure the displacement of the BB were standardized in all the volunteers as previously described in the section about the experimental procedure.

Transverse TAUS using on-screen calipers has shown to be a reliable method to measure the displacement of the BB during PFM contractions [5,6,7,8,9,10]. However, to our knowledge, no study has developed a customized code for this kind of measurement to date. The high intra-rater ICC values obtained during the requested PFM contractions (ICC > 0.90 for all four maneuvers) with the Ecolab code make this two-dimensional US tool a reliable method for measuring the displacement of the BB in women. 

### 4.3. Limitations of the Study

One of the limitations of this study was that transverse TAUS does not have a fixed reference point, unlike transperineal US, which is regarded as the gold standard for assessing bladder neck displacement in functional activities [22,24]. Since the BB displacement can only be expressed relative to a potentially mobile starting point, the transducer position needs to be consistent in order to achieve accurate and repeatable measurements. In line with this recommendation, Whittaker et al. [25] reported that, as long as the transducer motion is kept below approximately 5 to 10 degrees of angular motion or 10 mm of inward/outward motion, differences in measurements of the BB position are not statistically significant (*p* > 0.05). These findings provide guidance on acceptable amounts of transducer motion relative to the pelvis when recording measurements of BB displacement.

Another limitation of the study is that the inter-rater reliability could not be studied since only one rater was included. Therefore, studies with more than one examiner are needed in future research.

### 4.4. Clinical Implications

The implementation of this new algorithm in the investigation of the female pelvic floor can provide several advantages: the measurements do not need to be estimated during the data collection but can be performed offline without the patient’s presence; it is considerably less time consuming; the findings are more accurate; and the results are automatically saved in a proper datasheet for further analysis.

The MATLAB code designed by this research team to measure the displacement of the bladder base is available free as Appendix A to this article.

## 5. Conclusions

This two-dimensional US imaging method based on a custom MATLAB code could be a viable tool to measure offline the displacement of the BB in women during PFM contractions offline versus the manual measurement with calipers on the US screen. In addition, this method appeared to be highly reliable and therefore is potentially useful for further studies of the pelvic floor and abdominal contractions.

Based on the findings of the present study, we recommend the use of this MATLAB code in future studies to assess the immediate effect of functional activities on the displacement of BB. Further research is warranted to evaluate the potential clinical implication for the treatment and prevention of urogynecological dysfunctions in women.

## Figures and Tables

**Figure 1 jcm-11-02319-f001:**
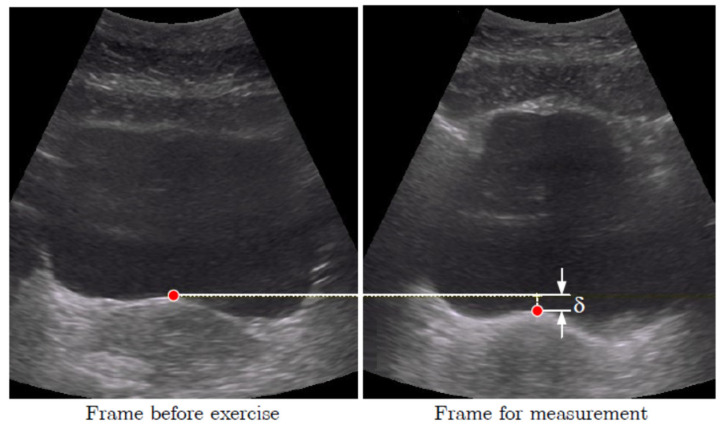
Placement of the marker in the middle of the bladder base. Displacement δ of the bladder base between the resting position (**left**) and the position during the contraction (**right**).

**Figure 2 jcm-11-02319-f002:**
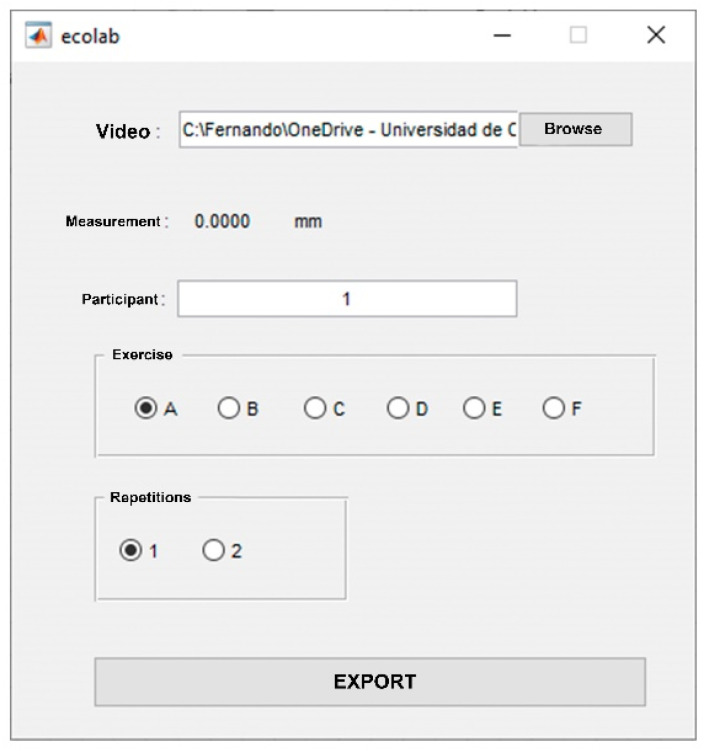
Graphical user interface.

**Figure 3 jcm-11-02319-f003:**
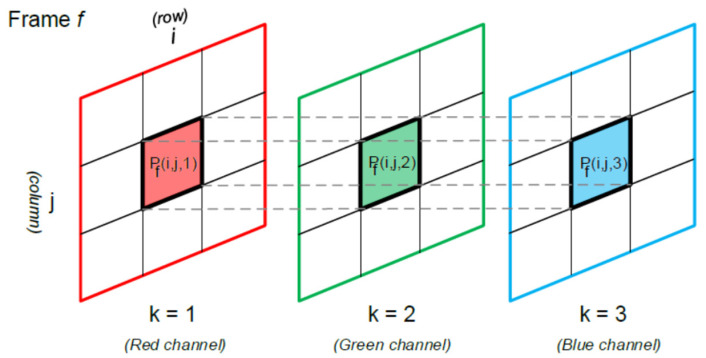
The value of $P_{f}\left(i,j,k\right)$ represents the color intensity of channel k of the pixel located in row *i* and column *j* of frame f.

**Figure 4 jcm-11-02319-f004:**
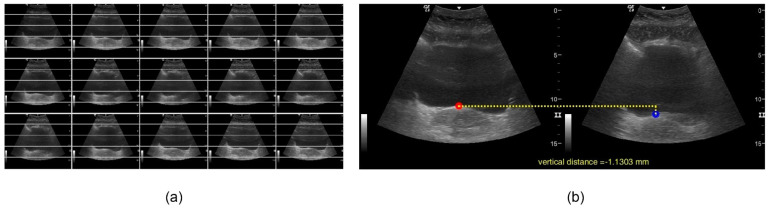
(**a**) Collage of frames. (**b**) Visualization of results.

**Figure 5 jcm-11-02319-f005:**
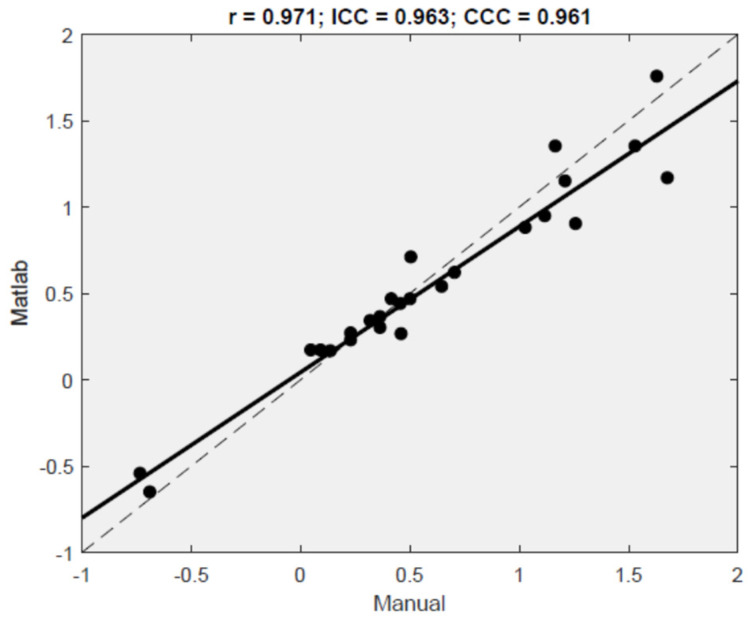
Scatterplot and correlation values for the measures given by Ecolab on the *y*-axis and by ultrasound transducer on the *x*-axis, with the linear regression fit (solid line) and the identity *y* = *x* line (dashed 45° line).

**Figure 6 jcm-11-02319-f006:**
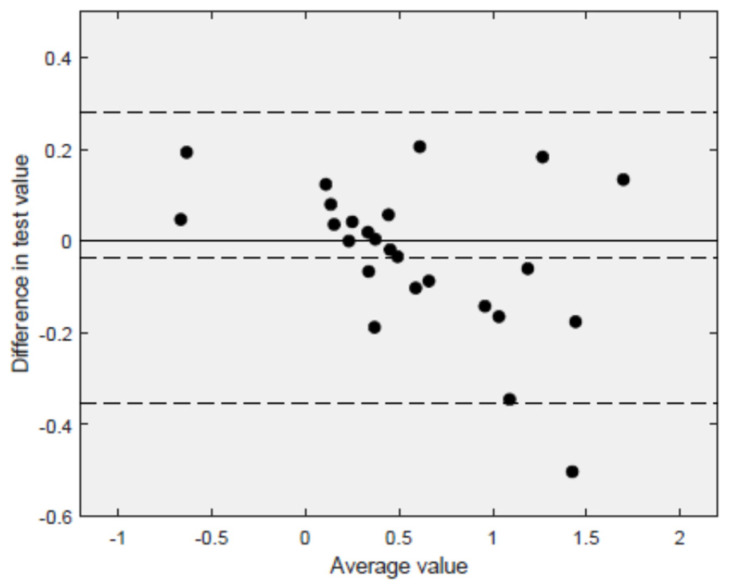
Bland–Altman plot (*n* = 27 volunteers). Differences in the measurements estimated with the Ecolab and the ultrasound transducers on the *y*-axis are plotted against the mean of the measurements with both methods on the *x*-axis. The mean difference ($\bar{d}=0.037$) and the relevant 95% confidence limits ($\bar{d}-1.96SD,\;\bar{d}+1.96SD)=\left(-0.353,\;0.279\right)$ are indicated by the horizontal dashed lines.

**Table 1 jcm-11-02319-t001:** Four perineal and abdominal contractions A–D that participants were instructed to perform randomly.

EXERCISES (PFM AND ABDOMINAL CONTRACTIONS A–D)	DESCRIPTION	PFM PRE-CONT ^a^	EST. TIMELINE ^b^ (S)
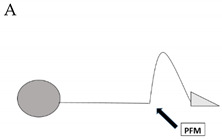	Submaximal isometric PFM contraction while breathing out	YES	3–7
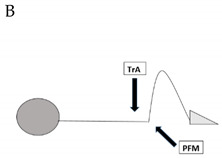	Submaximal isometric PFM and TrA ^c^ contraction while breathing out	YES	3–7
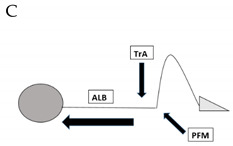	Equal to Contraction B + axial elongation of the whole spine (AEB) ^d^	YES	3–7
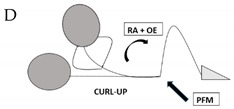	PFM submaximal contraction RA ^e^ + OE ^f^ OI ^g^ + holding apnea	YES	3–7

^a^ PFM PRE-CONT, Pelvic floor muscles pre-contraction held during the whole contraction; ^b^ EST TIMELINE, Estimated timeline in seconds; ^c^ TrA, Transversus abdominis muscle; ^d^ AEB, Axial elongation of the back; ^e^ RA, Rectus Abdominis; ^f^ OE, Obliquus externus muscle; ^g^ OI, Obliquus internus muscle.

**Table 2 jcm-11-02319-t002:** Validity of the MATLAB algorithm (Ecolab) to measure the displacement (cm) compared with the ultrasound transducer (manual) and inter-day reliability of the MATLAB algorithm.

Perineal and Abdominal Contraction	MATLAB Algorithm Validity (*n* = 27) Manual vs. Ecolab		MATLAB Algorithm Inter-Day Reliability (*n* = 32)	
	ICC (A,2) *n* = 27	95% CI	ICC (1,2) *n* = 32	95% CI
A	0.96	(0.92, 0.98)	0.96	(0.92, 0.98)
B			0.98	(0.97, 0.99)
C			0.99	(0.99, 0.99)
D			0.98	(0.97, 0.99)

ICC = intraclass correlation coefficient; CI = confidence interval.

## Data Availability

The data presented in this study are available on request from the corresponding author. The data are not publicly available due to ethical restrictions.

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
