# Peer review of "Validity and Reliability of an Offline Ultrasound Measurement of Bladder Base Displacement in Women"

_jcm, 2022, doi:10.3390/jcm11092319_

Round 1
Reviewer 1 Report
This is an exciting study using 2D US powered by a custom MATLAB code measuring bladder floor displacement in women during pelvic floor muscle contractions. My primary concern is the similarity to the previously published texts that the authors published - https://pubmed.ncbi.nlm.nih.gov/33871665/. I understand the method could be similar, but there is a commonly reused or recycling of own words mainly in the discussion part of the paper. I recommend revising the similarity of the paper
during revision.
Minor comments:
Abstract:
Line 19 – insert “the” before the words “bladder base”
Line 20 – replace “to” with “in” and replace “discriminate” with “determine”
Line 22 – missing a period
Line 26 – space after “ICC” and replace “checed (not a word)” with “confirmed”
Line 27 to 29 – Either make two sentences, or rephrase to avoid run on sentence
Introduction:
Line 37: colon after “used” instead of comma, replace “when” with “in which?
Line 46-49: should read “Advantages of using TAUS in the general exercising population include its high speed results, non-invasive technique, absence of the need to be undressed, and its direct visualization of pelvic floor movement during contractions.”
Line 52: delete he first “the” and replace “from” with”with”
Line 53: “during” is missing the letter d
Line 55: should read “…which could reduce the potential for measurement error when finding the displacement…”
Line 63: should be “descent” not “descend” also replace “could be not proper in” with “may not be advisable in”
Methods:
Line 98: replace “is required to be” with “was”
Table 1: first row second column has too many spaces after “PFM” Also some kind of diagram for the exercised would be helpful because I did not realize that the drawings on the left were people exercising until I read the explanation underneath
Line 126: Insert “The” at the beginning of the sentence and “the user” after “allows”
Line 166: the “(see Figure 4B)” should go after “Graphical User Interface” in the line before
Discussion:
Line 230: I am not sure what this is saying
Line 261: replace “has not” with “do not have”
Line 273: replace “another” with “further”
Line 279: delete “then”
Conclusions
Line 285: delete “valid” and insert “the” before “bladder”
Line 286: replace “…, comparing” with “when compared”
Author Response
Toledo (Spain), 08th April 2022
COVER LETTER. jcm-1658290
ATTN: Reviewer 1, Journal of Clinical Medicine.
Dear Reviewer,
Thank you very much for your revision of the manuscript titled “Validity and reliability of an offline ultrasound measurement of bladder base displacement in women”.
We really appreciate all the comments and recommendations. We would like to thank the reviewers for their effort in reviewing our work and furthermore for their kind suggestions that will help to improve the quality of our manuscript. We have tried to address all their concerns and comments.
Please find below the “List of replies” to your comments.
Yours sincerely.
Corresponding author: Asunción Ferri Morales, University of Castilla-La Mancha. Faculty of Physiotherapy and Nursing, Real Fábrica de Armas s/n. CP 45071, Toledo, Spain. Email: asuncion.ferri@uclm.es
LIST OF REPLIES TO REVIEWER 1
MANUSCRIPT ID: jcm-1658290
“Validity and reliability of an offline ultrasound measurement of bladder base displacement in women”.
Following here is a point-by-point response to each of the comments and a detailed description of the changes made in the manuscript.
*REVIEWER #1:
1-Primary comments
This is an exciting study using 2D US powered by a custom MATLAB code measuring bladder floor displacement in women during pelvic floor muscle contractions. My primary concern is the similarity to the previously published texts that the authors published - https://pubmed.ncbi.nlm.nih.gov/33871665/. I understand the method could be similar, but there is a commonly reused or recycling of own words mainly in the discussion part of the paper. I recommend revising the similarity of the paper during revision.
-Authors: Thank you for pointing this out. We have addressed this issue by rephrasing the paragraphs that were very similar to our previous publication. Please, check the revised manuscript with the changes tracked that has been uploaded to the Journal platform.
2-Minor comments:
Abstract:
Line 19 – insert “the” before the words “bladder base”
Line 20 – replace “to” with “in” and replace “discriminate” with “determine”
Line 22 – missing a period
Line 26 – space after “ICC” and replace “checed (not a word)” with “confirmed”
Line 27 to 29 – Either make two sentences, or rephrase to avoid run on sentence
-Authors: Your comments are highly appreciated. As suggested by the reviewer 1, we have corrected all these mistakes in the abstract. Please, check the revised manuscript with the changes tracked that has been uploaded to the Journal platform.
Introduction:
Line 37: colon after “used” instead of comma, replace “when” with “in which?
Line 46-49: should read “Advantages of using TAUS in the general exercising population include its high speed results, non-invasive technique, absence of the need to be undressed, and its direct visualization of pelvic floor movement during contractions.”
Line 52: delete he first “the” and replace “from” with”with”
Line 53: “during” is missing the letter d
Line 55: should read “…which could reduce the potential for measurement error when finding the displacement…”
Line 63: should be “descent” not “descend” also replace “could be not proper in” with “may not be advisable in”
-Authors: Your comments are highly appreciated. As suggested by the reviewer 1, we have corrected all these mistakes in the introduction. Please, check the revised manuscript with the changes tracked that has been uploaded to the Journal platform.
Methods:
Line 98: replace “is required to be” with “was”
Table 1: first row second column has too many spaces after “PFM” Also some kind of diagram for the exercised would be helpful because I did not realize that the drawings on the left were people exercising until I read the explanation underneath
Line 126: Insert “The” at the beginning of the sentence and “the user” after “allows”
Line 166: the “(see Figure 4B)” should go after “Graphical User Interface” in the line before
-Authors: Your comments are highly appreciated. As suggested by the reviewer 1, we have corrected all these mistakes in the methods section. Please, check the revised manuscript with the changes tracked that has been uploaded to the Journal platform.
Discussion:
Line 230: I am not sure what this is saying
Line 261: replace “has not” with “do not have”
Line 273: replace “another” with “further”
Line 279: delete “then”
-Authors: Your comments are highly appreciated. As suggested by the reviewer 1, we have corrected all these mistakes in the discussion section. Please, check the revised manuscript with the changes tracked that has been uploaded to the Journal platform.
Regarding LINE 230 “I am not sure what this is saying”, we would like to clarify that in statistics, the multiple correlation coefficient is a measure of how well a given variable can be predicted using a linear function of a set of other variables, in other words, it is the correlation between the variable values and the best predictions that can be computed linearly from the predictive variables.
So, in that paragraph of the manuscript, we tried to explain that the manual approach and Matlab algorithm have previously shown close agreement to record human body measurements. Although the study cited under reference 21 measured “the gastrocnemius fascicle length during gait” and not the bladder base displacement like we did, it could be an example of similar studies.
Conclusions
Line 285: delete “valid” and insert “the” before “bladder”
Line 286: replace “…, comparing” with “when compared”
-Authors: Your comments are highly appreciated. As suggested by the reviewer 1, we have corrected all these mistakes in the conclusions section. Please, check the revised manuscript with the changes tracked that has been uploaded to the Journal platform.
Reviewer 2 Report
This study aimed to introduce a 2-dimensional US algorithm (MATLAB algorithm) to measure offline the displacement of bladder base (BB) during pelvic floor muscles (PFM) contractions in women, and to analyze its validity and reliabilities. Total 32 nulliparous women were enrolled in this study. The study shows that MATLAB algorithm and ultrasound transducer measures did not differ statistically, both methods showed a very strong relationship, and the Ecolab demonstrated to be a valid and reliable method. The authors concluded that MATLAB algorithm seemed to be a valid and reliable tool to assess the effect of abdominal contractions in the female pelvic floor.
The basic rational of the study is good. There are several major and minor comments which I feel are important for the authors to address.
Major
1, The authors recruited total 32 volunteers, but they did not describe at the part of “PARTICIPANT” in Materials and Methods. The authors should address the other 5 volunteers in this part.
Minor
- There are many spelling mistakes and grammatical errors. The authors should be checked this article by native English speakers.
- (For examples, P1 L29 realiable, L34 visualise, L39 TPUS (no full spell)……., P6 L170 BD distance, so on)
- Figure 2 a) is not visible. The authors should change it.
Author Response
Toledo (Spain), 08th April 2022
COVER LETTER. jcm-1658290
ATTN: Reviewer 2, Journal of Clinical Medicine.
Dear Reviewer,
Thank you very much for your revision of the manuscript titled “Validity and reliability of an offline ultrasound measurement of bladder base displacement in women”.
We really appreciate all the comments and recommendations. We would like to thank the reviewers for their effort reviewing our work and furthermore for their kind suggestions that will help to improve the quality of our manuscript. We have tried to address all their concerns and comments.
Please find below the “List of replies” to your comments.
Yours sincerely.
Corresponding author: Asunción Ferri Morales, University of Castilla-La Mancha. Faculty of Physiotherapy and Nursing, Real Fábrica de Armas s/n. CP 45071, Toledo, Spain. Email: asuncion.ferri@uclm.es
LIST OF REPLIES TO REVIEWER 2
MANUSCRIPT ID: jcm-1658290
“Validity and reliability of an offline ultrasound measurement of bladder base displacement in women”.
Following here is a point-by-point response to each of the comments and a detailed description of the changes made in the manuscript.
*REVIEWER #2:
This study aimed to introduce a 2-dimensional US algorithm (MATLAB algorithm) to measure offline the displacement of bladder base (BB) during pelvic floor muscles (PFM) contractions in women, and to analyze its validity and reliabilities. Total 32 nulliparous women were enrolled in this study. The study shows that MATLAB algorithm and ultrasound transducer measures did not differ statistically, both methods showed a very strong relationship, and the Ecolab demonstrated to be a valid and reliable method. The authors concluded that MATLAB algorithm seemed to be a valid and reliable tool to assess the effect of abdominal contractions in the female pelvic floor.
The basic rational of the study is good. There are several major and minor comments which I feel are important for the authors to address.
1-Major comments
The authors recruited total 32 volunteers, but they did not describe at the part of “PARTICIPANT” in Materials and Methods. The authors should address the other 5 volunteers in this part.
-Authors: Your comment is highly appreciated. As suggested by the reviewer, we have added information about the subjects in the METHODOLOGY SECTION-PARTICIPANTS.
Please, check that the “Participant” paragraph has been rewritten (lines 80-83) as follows:
“A convenience sample of 32 nulliparous women participated in this prospective study, 27 to calculate the validity of the MATLAB algorithm and 5 additional volunteers to determine its reliability..”
2-Minor comments
There are many spelling mistakes and grammatical errors. The authors should be checked this article by native English speakers.
(For examples, P1 L29 realiable, L34 visualise, L39 TPUS (no full spell)……., P6 L170 BD distance, so on).
-Authors: Thank you for your valuable comment and examples about the spelling mistakes and grammatical errors.
The manuscript has been sent to a professional editing company to improve the spelling and grammar. Please, check the revised manuscript submitted to the journal platform.
Figure 2 a) is not visible. The authors should change it.
-Authors: Thank you for pointing this out.
We have removed the FIGURE 2a) since we thought that it did not provide any relevant information to the readers.
Reviewer 3 Report
Thank you for the opportunity to review this manuscript
The authors have made considerable efforts to develop this paper, however, I believe that the current version of manuscript should be improved through some minor and re-writing. I want to provide some suggestions for the improvement of this paper as follows.
I recommend to highlight the novelty of your study according with previous researches.
What were inclusion and exclusion criteria of sample.
The Conclusions section need to be revised and extending with more relevant ideas related your main findings.
Author Response
COVER LETTER. jcm-1658290
ATTN: Reviewer 3, Journal of Clinical Medicine.
Dear Reviewer,
Thank you very much for your revision of the manuscript titled “Validity and reliability of an offline ultrasound measurement of bladder base displacement in women”.
We really appreciate all the comments and recommendations. We would like to thank the reviewers for their effort in reviewing our work and furthermore for their kind suggestions that will help to improve the quality of our manuscript. We have tried to address all their concerns and comments.
Please find below the “List of replies” to your comments.
Yours sincerely.
Corresponding author: Asunción Ferri Morales, University of Castilla-La Mancha. Faculty of Physiotherapy and Nursing, Real Fábrica de Armas s/n. CP 45071, Toledo, Spain. Email: asuncion.ferri@uclm.es
LIST OF REPLIES TO REVIEWER 3
MANUSCRIPT ID: jcm-1658290
“Validity and reliability of an offline ultrasound measurement of bladder base displacement in women”.
Following here is a point-by-point response to each of the comments and a detailed description of the changes made in the manuscript.
*REVIEWER #3:
Thank you for the opportunity to review this manuscript
The authors have made considerable efforts to develop this paper, however, I believe that the current version of manuscript should be improved through some minor and re-writing. I want to provide some suggestions for the improvement of this paper as follows.
First comment
I recommend to highlight the novelty of your study according with previous researches.
-Authors: Your comment is highly appreciated. As suggested by the reviewer, we have added information to highlight the novelty of our study in the second paragraph of the “DISCUSSION section” (lines 654-655):
“However, the Ecolab represents a novel means to measure bladder base displacement during different exercises or functional activities.”
Second comment
What were inclusion and exclusion criteria of sample.
-Authors: Thank you for your comment. Please, find the “Inclusion and Exclusion criteria of sample” at the first paragraph of the “Materials and Methods Section-PARTICIPANTS” (lines 82-88):
“The inclusion criteria were to be nulliparous, willingness to participate in the study and ability to contract PFM correctly. This ability was assessed by palpation and by superficial biofeedback electromyography (PHENIX® USB NEO, Vivaltis. Montpellier, FR), reflecting the intensity and the length of the pelvic floor contraction in a monitor screen. Exclusion criteria were inability to contract PFM properly, pregnancy, known neurological disease, or inability to understand instructions given in Spanish language.”
Third comment
The Conclusions section need to be revised and extending with more relevant ideas related your main findings.
-Authors: Thank you for your comment. As suggested by the reviewer, we have added one paragraph at the end of the CONCLUSIONS section (lines 922-925):
“Based on the findings of the present study, we recommend the use of this MATLAB code in future studies to assess the immediate effect of functional activities on the displacement of BB. Further research is warranted to evaluate the potential clinical implication for the treatment and prevention of urogynecological dysfunctions in women.”
Round 2
Reviewer 1 Report
The authors responded well to all my comments.